# Etiology of Exercise Injuries in Firefighters: A Healthcare Practitioners’ Perspective

**DOI:** 10.3390/healthcare11222989

**Published:** 2023-11-19

**Authors:** Alyssa Q. Eastman, Beth Rous, Emily L. Langford, Anne Louise Tatro, Nicholas R. Heebner, Phillip A. Gribble, Rosie Lanphere, Mark G. Abel

**Affiliations:** 1First Responder Research Laboratory, University of Kentucky, Lexington, KY 40506, USA; 2Department of Kinesiology and Health Promotion, University of Kentucky, Lexington, KY 40506, USA; rosie.lanphere@uky.edu; 3Department of Clinical Genomics Research, Mayo Clinic, Rochester, MN 55905, USA; 4Department of Educational Leadership Studies, University of Kentucky, Lexington, KY 40508, USA; beth.rous@uky.edu; 5Department of Health and Human Sciences, University of Montevallo, Montevallo, AL 35115, USA; 6Sports Medicine Research Institute, University of Kentucky, Lexington, KY 40536, USA; 7Department of Athletic Training and Clinical Nutrition, University of Kentucky, Lexington, KY 40536, USA

**Keywords:** firefighter, musculoskeletal injury, healthcare practitioner, risk factor, exercise

## Abstract

The purpose of this study was to query healthcare practitioners (HCPs) who treat firefighter injuries to identify risk factors and mechanisms associated with musculoskeletal injuries during exercise. A phenomenological design was utilized to understand the experiences of HCPs while treating firefighters’ musculoskeletal injuries due to exercise. Semi-structured interviews were conducted with 14 HCPs. Two interviews were pilot-tested with HCPs to ensure reliability and validity. Interviews were transcribed and uploaded to a qualitative analysis software program. Although the study inquired about injuries incurred by any exercise modality (e.g., endurance and resistance training), injuries induced during resistance training were prominent among HCPs as resistance training emerged as a primary exercise injury mechanism. HCPs indicated that the back and shoulder were prevalent anatomical exercise injury locations. Risk factors for exercise injuries included age, immobility, movement proficiency, and factors associated with fatigue. Exercise injury mechanisms included poor resistance training technique and overexertion. These findings could guide exercise program design, use of movement assessments, and the identification of other countermeasures to decrease the risk of resistance training exercise injuries among firefighters.

## 1. Introduction

Firefighting requires the performance of vigorous occupational tasks [1]. It is critical that firefighters engage in regular exercise to enhance their health, occupational readiness, and safety. Despite concerns that performing on-duty exercise may reduce subsequent occupational readiness due to fatigue [2], cross-sectional research indicates that regular on-duty exercise likely facilitates a greater work capacity, even in a post-exercise fatigued state compared with a sedentary non-fatigued state [3]. None the less, regarding health, sudden cardiac events are among the leading causes of on-duty fatality among firefighters [4]. Indeed, the prevalence of cardiovascular disease risk factors among firefighters has been extensively described in the literature [5,6]. Regular exercise and greater physical fitness levels can attenuate many of these modifiable risk factors [5]. Regarding occupational readiness, various biomotor abilities are required to effectively perform fireground tasks [7,8,9]. As such, exercise interventions have been demonstrated to enhance relevant biomotor abilities and occupational physical abilities in structural firefighters [10,11,12]. Regarding safety, firefighting is a dangerous occupation, as the injury rates are greater than those in law enforcement and other public and private sector occupations with similar demands [13]. However, research has indicated that physical fitness is related to the incidence of firefighters’ occupational injuries [14]. Collectively, exercise is a critical countermeasure to enhance firefighters’ health, occupational readiness, and safety.

Ironically, despite the benefits of exercise for firefighters, exercise is among the leading causes of occupational injuries, accounting for about 27–33% of all injuries [15,16]. Poplin et al. reported that 85% of exercise injuries were musculoskeletal in nature, classified as sprains and strains, and primarily induced by acute overexertion [16]. Similarly, Frost et al. found that 27% of all sprain and strain injuries were induced during physical training [17]. These injuries occurred most frequently at the shoulder (32%), ankle (30%), knee (22%), and back (18%) [17]. In addition, these musculoskeletal injuries are associated with substantial fiscal and personnel consequences. For instance, musculoskeletal injuries have been reported to cost, on average, USD 8031 per incident in medical and worker compensation costs [18]. These costs do not account for the expenses associated with backfilling the injured firefighter’s line position. Interestingly, despite the risk of exercise injury, firefighters that regularly exercise are half as likely to sustain a non-exercise occupational injury [15], potentially demonstrating the importance of exercise in occupational injury mitigation.

The existing data describe the prevalence and type of exercise injuries in the Fire Service. However, these surveillance and survey-based data lack detail and context regarding exercise injury mechanisms and risk factors. Obtaining additional detail and context could guide the development of targeted exercise injury mitigation interventions. Healthcare practitioners, such as athletic trainers and physical therapists, who regularly treat firefighters’ exercise injuries are uniquely positioned to provide valuable insight to identify contextual factors surrounding these risk factors and mechanisms for injury. These healthcare practitioners have the educational training to identify etiological factors (i.e., intrapersonal, interpersonal, and institutional) associated with injury occurrence and understand the occupational and exercise environments within the Fire Service. Therefore, the aims of this study were threefold: (1) to describe prevalent injury types and anatomical locations of exercise injuries; (2) to identify common intrapersonal, interpersonal, and institutional risk factors of exercise injuries; and (3) to identify the mechanisms of exercise injuries from the clinical experiences of healthcare practitioners who regularly treat firefighter injuries. We hypothesized that healthcare practitioners would identify sprains and strains to be the most prevalent types of injuries, and that anatomical injuries would typically be incurred at the shoulder, back, and knee, and that resistance training (i.e., lifting) mechanics would be identified as a common reason for exercise injuries.

## 2. Materials and Methods

### 2.1. Research Design

This study utilized a phenomenological approach involving semi-structured interviews with healthcare practitioners who treat firefighter injuries. Phenomenology is a qualitative methodology that is used to understand shared experiences within a specific group [19]. Semi-structured interviews are well suited to obtain in-depth perspectives around a phenomenon of interest. This method has been previously implemented in studies exploring firefighters’ lived experiences [20,21,22].

### 2.2. Participants

The study was conducted according to the guidelines of the Declaration of Helsinki and approved by the University’s Institutional Review Board (Protocol# 74158, Approval: 12/07/2021). Informed consent was obtained from all participants involved in the study prior to participation. Using convenience sampling, 14 licensed healthcare practitioners were recruited who had experience treating firefighter injuries within the past year. Practitioners had an average of 10.4 years of clinical experience, including 4.4 years with tactical populations (Table 1). Collectively, they reported approximately 89,000 total encounters (encounters = weekly patient load × 52 week/year × years of experience) treating firefighter injuries.

Practitioners were nationally recruited through professional networks and fire departments with embedded healthcare. Inclusion criteria required ≥1 year of experience treating firefighter injuries and current clinical practice. Practitioners included athletic trainers, physical therapists, physicians, and nurse practitioners.

### 2.3. Procedures

Semi-structured interviews lasting approximately 45 min were conducted with a sample of 14 healthcare practitioners. The number of participants included in the sample was the result of investigators obtaining data saturation for the study aims. The interview protocol was drafted in consultation with an external healthcare practitioner who regularly treats tactical populations. Pilot interviews were conducted with two healthcare practitioners (excluded from data analysis) to provide feedback and guide revisions to the interview script. Interviews were conducted using video conference technology (Zoom, Version 5.5.2, San Jose, CA, USA). Conrad and colleagues’ conceptual framework was applied and utilized to develop the script and deductive matrix for the current study [23]. This framework described the inter-relationships of personal worker, workplace, and environmental factors on the risk of musculoskeletal injuries [23]. The interview protocol included four sections. Section 1 outlined the purpose of the study, obtained verbal informed consent, addressed issues of confidentiality, and established trust with healthcare practitioners. Section 2 included the interview questions that explored the healthcare practitioners’ experience in treating and rehabilitating firefighter injuries. Section 3 included open-ended questions related to injury mechanisms and environmental factors associated with exercise. Follow-up questions and probes were included as needed. For the context of this study, exercise injuries were operationally defined as an injury incurred due to direct participation in any exercise modality for which healthcare treatment was sought. Section 4 was composed of closing questions that allowed the participants to add additional information not covered in other sections.

### 2.4. Data Analysis

Interviews were recorded, transcribed verbatim, and analyzed using Dedoose software (Version 8.0.35, SocioCultural Research Consultants, LLC, Manhattan Beach, CA, USA). A deductive coding framework was developed based on the research aims and Conrad et al.’s injury determinants model, which included five major areas (i.e., main codes): injury type, location, setting, risk factors, and mechanisms (Table 2) [23]. In addition to deductive coding, inductive analysis further categorized specific details from the main codes into sub-codes (Table 3). A code co-occurrence analysis examined the intersections between main and sub-codes. Strategies to establish data trustworthiness included analyst triangulation, members checking transcripts with participants, and maintaining an audit trail of analytical decisions.

Two researchers independently coded the transcripts through multiple rounds of iterative analysis. After initial coding, the researchers discussed inconsistencies, clarified code definitions, and refined the codebook. Interrater reliability was assessed using Cohen’s kappa on two fully coded transcripts. A kappa of 0.81 indicated a high level of coding agreement.

## 3. Results

### 3.1. Injury Locations and Types

The first aim of the study was to identify the most prevalent exercise injury locations and types. To address this question, 18 excerpts of data were used for the analysis. The shoulder and back were the most prevalent injury locations, with chronic-type sprains and strains as the most common injury types (Table 4). As one practitioner stated, “During exercise, I’d say for the most part they are chronic, so if these are more non-line of duty, I was doing this on my own, they are chronic. Not even tendonitis, but almost tendinosis that we’re mainly seeing because they aren’t the type of individuals to seek immediate acute care. You know, they kind of work through a lot of things they’re used to being uncomfortable every day and what’s an ache or pain? You know? So, we don’t really catch those until they’ve been going on for quite a while” (Participant: P2).

### 3.2. Risk Factors

The second aim of the study was to identify intrapersonal, interpersonal, and institutional risk factors. For this aim, 73 excerpts related to exercise injury risk factors were analyzed. Five key risk factor themes emerged: age, immobility, movement proficiency, fatigue/recovery, and mental health (Table 5).

Age was a complex intrapersonal risk factor. Some practitioners noted that younger firefighters tended to be more injury-prone due to competitiveness and overexertion. As one stated, “The younger individuals I’ve seen...it sounds more related to fatigue” (P12). Others cited age-related wear-and-tear in older firefighters. The influence of age also depended on the practitioner’s experience in the department. That is, more experienced healthcare practitioners tended to express that the younger firefighters were more likely to sustain an exercise-related injury compared with older firefighters. 

Immobility and muscle tightness were frequently noted as precursors to exercise injury through improper mechanics, as highlighted by one practitioner: “Immobility lends itself to improper technique and improper movement, and then you’ve end up with either an injury there or compensatory injury” (P7). Poor movement proficiency was consistently tied to improper lifting technique. For instance, one practitioner discussed correcting improper lifts during exercise in order to improve occupational movement proficiency, “Yeah, we’ve done a lot with retraining specifically with like squat and deadlift [exercises] and then trying to correct those improper movements and alleviate glute tightness or glute weakness so that things that are triggering the way that they should be, and proper movement is engaged” (P7).

Fatigue, lack of recovery and sleep were cited as multifaceted risk factors requiring a holistic approach to manage. Secondary labor jobs contributed to overall fatigue concerns. As one practitioner said, “They kind of work through a lot of things they’re used to being uncomfortable every day” (P2). 

Mental health also had a complex relationship with exercise injury risk. Chronic psychological stress was seen as a factor in potential injury susceptibility during exercise, whereas exercise was also described as a psychological coping mechanism. Peer support resources were also cited as important to address firefighters’ mental stress. 

### 3.3. Injury Mechanisms

The third aim of this study focused on identifying the mechanisms of exercise injuries among firefighters. To address this aim, 35 excerpts were identified and used for analysis. Poor exercise technique was the most prevalent mechanism, especially with resistance training (Table 6). Health care practitioners described that a lack of education regarding proper technique contributes to injury. One practitioner tied exercise technique and age together: “…a lot of the older guys it’s [injuries are] more from poor technique. Whereas some of the younger individuals I’ve seen… it sounds more related to fatigue. So, whether that’s related to technique or overloading the bar, I don’t know. But I would say more related to technique” (P12). Furthermore, overexertion during resistance training was also cited. General movement deficiencies interacting with other risks were discussed as well.

Several practitioners agreed that regardless of exercise type, injuries tended to include the mechanism of overexertion. One practitioner tied these findings together in one response, “I think during exercise it might be overexertion or like I would say, like an exacerbation of something that’s chronic, like someone who had maybe an old injury that they’ve never treated, and it kind of gets exacerbated with like a little bit of too intense exercise coupled with like a poor movement pattern. Maybe they haven’t slept well, they’re not moving well, kind of like a combination of things and work or training related…” (P4). One practitioner indicated that years of exposure to high-intensity-interval-style training may have led to overexertion and overuse injuries (P9). 

## 4. Discussion

The first aim of this study was to describe the practitioners’ perspective regarding firefighters’ exercise injury location and type. Practitioners indicated that the shoulder and back were the most prevalent anatomical locations of injury, followed by the knee, ankle, and elbow. Similarly, Frost et al. reported that the most prevalent exercise injury locations were the shoulder (22%), ankle (16%), knee (11%), and back (9%) [17]. Furthermore, practitioners in the present study indicated that exercise injuries tended to be chronic in nature (53% of excerpts), whereas acute injuries were less prevalent (29% of excerpts). In addition, sprain/strain was the most reported injury type (65%). These findings supported previous work by Frost et al. and Orr et al., who reported that sprain/strain was the primary cause of all musculoskeletal injuries in firefighters [17,24]. 

The second aim of this study was to identify the intrapersonal, interpersonal, and institutional risk factors for exercise injuries among firefighters. The practitioners indicated that the type of injury incurred may have been associated with firefighters’ age, which may have had an influence on interpersonal and intrapersonal factors. Regarding the interpersonal context, practitioners indicated that some firefighters were competitive in nature and the younger firefighters often wanted to prove themselves. This competitive nature may have contributed to a greater risk of sustaining an exercise injury due to overexertion. Specifically, one healthcare practitioner explained that younger firefighters may have competed during exercise and that the competitive environment influenced overloading the bar (P7). Another practitioner described that some older firefighters had invested time, engrained proper exercise movement patterns, and addressed musculoskeletal deficits, which may have decreased their risk of exercise injuries compared with younger firefighters (P1). Regarding the intrapersonal context, one practitioner discussed the necessity of modifying exercise selection due to aging, indicating that after so many years of wear-and-tear, the cartilage in the shoulder was gone. Therefore, bench pressing was no longer feasible, and the type of exercise needed to change (P3). Interestingly, age has been shown to not impact the rate of neuromuscular recovery following a bout of on-duty resistance training between younger and older firefighters [25], thus older firefighters may not be at an increased risk of subsequent occupational injury due to fatigue or a lack of recovery. Contrarily, other physiological research has demonstrated that aging is associated with a decrease in musculoskeletal mass, strength, motor performance, flexibility, and ultimately, injury resilience [26]. 

Immobility and movement proficiency were prevalent intrapersonal injury risk factors reported by the healthcare practitioners and were often collectively described in their role in exercise injuries. Specifically, the healthcare practitioners discussed that immobility often attenuated movement proficiency (i.e., the ability to correctly perform exercises). This underscored the importance of incorporating proper movement education for firefighters during an exercise program. In support of this assertion, Frost et al. conducted a longitudinal training intervention among two groups of firefighters that were provided with either a combined fitness training program plus movement education or a fitness training program only [27]. The findings demonstrated that the combined group had improved movement proficiency on occupationally relevant tasks and enhanced physical fitness, whereas the training-only group had decreased movement proficiency following the training intervention. These results indicated that training without an awareness of movement quality may increase the risk of exercise and/or occupational injury. 

The practitioners indicated that enhanced physical fitness decreased the risk of exercise-related musculoskeletal injuries. Similarly, Poplin and colleagues demonstrated that firefighters with the lowest aerobic capacity (<43 mL/kg/min) were 2.5 times more likely to incur an exercise injury compared with those with the highest aerobic capacity (>48 mL/kg/min) [14]. The impact of physical fitness on occupational injuries is also supported in the literature. For instance, Butler et al. reported that aerobic fitness, the deep squat and push-up components of the Functional Movement Screening, and the sit-and-reach assessments were correlated to injury occurrence in an academy setting [28]. Likewise, Conrad et al. conducted focus groups with firefighters and fire chiefs and found that the level of physical fitness was a personal factor that was related to the incidence of occupational musculoskeletal injuries [23]. Furthermore, Jahnke et al. implemented an injury surveillance survey and found that those who exercised were 4.6 times more likely to sustain an exercise injury compared with those who did not exercise on-duty [15]. However, those who reported exercising on-duty were approximately half as likely to sustain an occupational injury. Thus, an adequate level of physical fitness is crucial for performance and minimizing injury risk; however, the current study revealed the importance of utilizing proper movement patterns while developing physical fitness. 

The intrapersonal risk factors of fatigue, recovery, and sleep were often described as overall health and wellness by the healthcare practitioners. One practitioner even stated that they believed that fatigue, recovery, and sleep were the biggest risk factors for sustaining a musculoskeletal injury (P3). However, the healthcare practitioners’ experiences differed regarding the sleep factor. That is, one healthcare practitioner indicated that sleep was not as significant as other risk factors because firefighters, in their experience, were used to a lack of sleep, and that it more so contributed to recovery and not being able to rest (P2). However, another healthcare practitioner described how detrimental fragmented sleep was with disrupted REM cycles and increases in cortisol levels that contributed to chronic stress (P11). However, all the practitioners agreed that the lack of recovery and fatigue impacted the risk of sustaining a musculoskeletal exercise injury. In support of this assertion, Gerstner and colleagues demonstrated that firefighters’ rate of force development (in 50 ms) decreased over several shift cycles [29], which may have been associated with neuromuscular fatigue and risk of slip, trip, or fall injuries. 

Interestingly, the intrapersonal risk factor of mental health only occurred in 15% of the excerpts of exercise injuries. However, Conrad et al. identified chronic stress, which encompassed mental, physical, and emotional health, as a risk factor for occupational musculoskeletal injury [23]. In juxtaposition, health care practitioners described that exercise could often be used as a coping mechanism for chronic stress and may have caused stress if firefighters became injured and were not able to exercise. This finding was supported by the literature, which has suggested that firefighters with worse self-reported mental quality of life were fitter in some regards [30]. Furthermore, a seasoned healthcare practitioner described the importance of mind, body, and spirit when discussing mental health and dealing with chronic stress. 

The third aim of this study was to identify potential mechanisms of exercise injuries. Two of the primary exercise injury mechanisms were poor resistance training technique and movement mechanics. These mechanisms were commonly associated with immobility and movement proficiency for resistance exercises. For instance, one healthcare practitioner discussed how the inadequate use of core bracing may have influenced injury risk. Other practitioners indicated that firefighters are generally poor movers. To that end, several healthcare practitioners discussed using movement screenings to assess overall movement quality and discussed how they had been able to predict injuries in firefighters. However, some healthcare practitioners discussed that they had developed their own method for assessing movement and did not utilize the Functional Movement Screen^©^. Several articles have discussed the efficacy of the Functional Movement Screen^©^ with firefighters and have found that it can provide general information regarding movement, but a specific screening is needed [31]. 

Overexertion was the final exercise injury mechanism identified by the healthcare practitioners. One practitioner indicated that chronic participation in ballistic or metabolic conditioning workouts may have been associated with overexertion and/or overuse injuries. Overexertion exercise injuries may be due to utilizing an acute or chronic resistance or endurance training load that exceeds the tissue’s capacity, which may be supported by the Overexertion Theory of injury [32]. Although difficult to ascertain, genetics, morphological, and psychosocial factors may play a mediating role in the exertion–tolerance relationship [32]. Interestingly, overexertion has also been cited as a primary cause of fireground injuries [18,24]. Thus, it is plausible that overexertion occupational injuries may be reduced if tissue tolerance is increased through movement pattern and energy-system-specific exercise training [33]. 

There were several limitations to the present study. First, the population recruited for the present study was composed of a convenience sample inclusive of a select number of healthcare practitioners who treated firefighters’ musculoskeletal injuries. This limited sample had vastly different roles and experiences. For example, the healthcare practitioners may have served in roles as athletic trainers, physical therapists, strength and conditioning coaches, or nurse practitioners. In addition, the healthcare practitioners’ connection to the Fire Service differed, as they may have been directly embedded within the department, contracted by the fire department through a company, or employed by the city. Secondly, the number of weekly patients differed between the providers. For example, one practitioner had 60+ weekly patient visits, whereas another practitioner discussed having two weekly patient visits. Finally, only a select few healthcare practitioners witnessed firefighters exercising and sustaining an injury. Other providers relied on information from the patient about the injury at the time of injury.

### Clinical Significance

The present study indicated that shoulder and back injuries commonly occurred during exercise in structural firefighters. These injuries tended to be preceded by tissue tightness, subsequently leading to joint immobility and poor movement proficiency during exercise. Tactical strength and conditioning practitioners and departmental healthcare practitioners should regularly evaluate firefighters’ mobility, especially in the thoracic spine and glenohumeral joint. Stability of the spinal erectors and scapulae should also be addressed as part of a comprehensive exercise program. Given the reported risk factor of fatigue, recovery, and sleep, practitioners should utilize a subjective (e.g., wellness questionnaire) or objective (e.g., rate of force development–vertical jump) daily monitoring tool to assess the physiological readiness to train and adapt to the training stimuli. This information should be used to identify trends in physiological readiness and intervene when appropriate. In addition, practitioners should educate firefighters about proper sleep hygiene to enhance sleep quantity and quality. For instance, caffeine use should be limited, perform regular exercise, attempt to maintain a regular sleep schedule on- and off-duty, and avoid screentime prior to sleeping. Fire departments should enhance sleep outcomes by providing resources, including independent sleeping quarters and emergency tones that only wake the appropriate personnel. Finally, given the reported exercise injury mechanism of resistance training technique, qualified strength and conditioning practitioners should be utilized to teach and ensure proper mechanics are used to decrease the risk of exercise injury. Enhancing firefighters’ movement proficiency during resistance training may transfer to use of proper movement patterns on the fireground. 

## 5. Conclusions

In conclusion, the present study indicated that the most prevalent exercise injuries were chronic in nature and affected the lower back and shoulder. Prevalent risk factors included age, immobility, and poor movement proficiency, whereas the mechanisms of injury included poor resistance training mechanics. Collectively, these findings provide tactical strength and conditioning practitioners, healthcare practitioners, and agency leadership with an etiological framework to develop appropriate countermeasures.

## Figures and Tables

**Table 1 healthcare-11-02989-t001:** Summary of healthcare practitioners’ demographic information and descriptive characteristics.

Subject #	Education	Clinical Credential	Years of Clinical Experience	Years of Experience with Tactical Population	Observed Firefighter Tasks	Firefighter Patient Encounters (Weekly)	Total Treatment Encounters
1	Bachelor’s	AT, CSCS	15	12	Yes	25	15,600
2	Doctorate	AT, CSCS	10	2	Yes	45	4680
3	Doctorate	AT	5	5	Yes	80	20,800
4	Master’s	AT	7	3	Yes	30	4680
5	Doctorate	PT, CSCS	10	8	Yes	8	3328
6	Doctorate	AT and PT	11	4	Yes	5	1040
7	Master’s	AT	17	7	Yes	60	21,840
8	Doctorate	PT	5	5	Yes	2	520
9	Doctorate	PT	13	7	Yes	25	9100
10	Master’s	AT	5	2	Yes	12	1248
11	Doctorate	NP	42	16	Yes	8	6656
12	Master’s	AT	3	1	Yes	-	-
Mean			11.9	6.0		27.3	8135.6
SD			10.4	4.4		25.1	7809.1

AT: Athletic Trainer; CSCS: Certified Strength and Conditioning Specialist; NP: Advanced Family Nurse Practitioner; PT: Physical Therapist.

**Table 2 healthcare-11-02989-t002:** Deductive coding framework for each study aim.

Main Code	Aim 1: Exercise Injuries	Aim 2: Exercise Injury Risk Factors	Aim 3: Exercise Injury Mechanisms
Injury type	X		
Injury location	X		
Injury setting	X	X	X
Risk factor		X	
Injury mechanism			X

**Table 3 healthcare-11-02989-t003:** Exercise injury categories by area of interest.

Main Code	Sub-Code
Injury Type	Acute
	Chronic
	Tendonitis/tendinosis
	Sprain/strain
	Chronic to acute
Injury Location	Ankle
	Back
	Elbow
	Knee
	Shoulder
Injury Setting	Exercise related
	Occupationally related
	Personal
Risk Factor	Biological age
	Fatigue
	Immobility
	Recovery
	Mental health
	Movement proficiency
	Training age
	Nutrition
Injury Mechanisms	Resistance training (poor lifting technique)
	Movement mechanics
	Overexertion

**Table 4 healthcare-11-02989-t004:** Prevalence of firefighter injuries occurring during exercise as described by 12 healthcare practitioners, stratified by anatomical location and type of injury.

Injury Anatomical Locations (18 Excerpts)				
Shoulder	Back	Knee	Ankle	Elbow
83%	73%	28%	17%	11%
**Type of Injury (17 Excerpts)**				
Sprains/Strains		Chronic	Acute	
65%		53%	29%	

**Table 5 healthcare-11-02989-t005:** Prevalence of healthcare practitioners’ reported risk factors of exercise injuries for structural firefighters.

Risk Factors	73 Excerpts
Biological age	21%
Fatigue, recovery, and sleep	21%
Immobility	19%
Mental health	15%
Movement proficiency	21%

**Table 6 healthcare-11-02989-t006:** Prevalence of healthcare practitioners’ reported exercise injury mechanisms for structural firefighters.

Injury Mechanism	35 Excerpts
Resistance training	51%
a. Poor resistance training technique	44%
Overexertion	20%
Movement mechanics	14%

## Data Availability

Restrictions apply to the availability of these data. Data were obtained directly from participants and are restricted from sharing with non-research personnel due to the Institution’s Human Subjects Protection regulations.

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
