# Peer review of "Etiology of Exercise Injuries in Firefighters: A Healthcare Practitioners’ Perspective"

_healthcare, 2023, doi:10.3390/healthcare11222989_

Round 1

Reviewer 1 Report

Comments and Suggestions for Authors

General Comments:

The study interviewed health care practitioners working with firefighters to address injuries to gain insight as to factors and mechanisms related to exercise related injuries. This area of work has value for the firefighting community as this topic (injuries caused by exercise in firefighters) is understudied, at least underreported in the existing literature. The findings are notable and will serve as a ‘lead-up research’ for future work in this area.  The manuscript was well-written. One broad comment for the manuscript is that the term ‘exercise injury’ is used; however, the focus appears to be narrower to a sub-domain of exercise – resistance training. Aerobic capacity is mentioned in the discussion. Adding some clarity and commentary around this issue would be a primary point of feedback.

Specific Comments

Abstract

-Earlier in the abstract should make the point that the study was focused on exercise injuries. Suggest editing line 16-17 to be: “….musculoskeletal injuries due to exercise.”

Introduction

- Adding a line (or two) to provide barriers to firefighters engaging in regular exercise would be of value. Several useful references are:

Lovejoy, S.; Gillespie, G.L.; Christianson, J. Exploring Physical Health in a Sample of Firefighters. Workplace Health Saf 2015, 63, 253–258, doi:10.1177/2165079915576922. 

Dennison, K.J.; Mullineaux, D.R.; Yates, J.W.; Abel, M.G. The Effect of Fatigue and Training Status on Firefighter Performance. J Strength Cond Res 2012, 26, 1101–1109, doi:10.1519/JSC.0b013e31822dd027.

-An operational definition of an ‘exercise injury’ would be useful.

Methods

-The methodology was overall sound but several questions arise: 1) how was the final number of participants determined? And 2) were subgroup analyses considered? From table 1 potential subgroups would be AT vs. PT and volume of FF patient encounters  as some were much higher than others (25+ per week vs. 12 or fewer).

Results

-The quotes are particularly interesting findings, in my perspective. Consider adding a table, perhaps supplementary table, to present direct quotes organized by themes.

Discussion

-Line 228-229 – I believe the authors mean “…..overexertion due to exercise.” Please specify if due to exercise or more broadly, meaning all aspects of their occupational duties.

-Paragraph 288-295 – The point about firefighters may use exercise as a coping mechanism is interesting and there is some support in the literature. A recent study reported that firefighters with worse self-reported mental quality of life were fitter in some regards.

Effects of Fitness on Self-Reported Physical and Mental Quality of Life in Professional Firefighters: An Exploratory Study. Work 2023, doi:10.3233/WOR-220673.

Clinical significance – several of the statements can be supported with references used earlier in the manuscript.

References

-# 5 and 12 is incomplete

Comments on the Quality of English Language

The quality of the english language was high.

Reviewer 2 Report

Comments and Suggestions for Authors

General Comments:

Thank you for this review. The manuscript was well-written. 

This area is valuable for the firefighting community as this topic (injuries caused by exercise in firefighters) is understudied. 

One broad comment for the manuscript is that the term ‘exercise injury’ is used; however, the focus appears to on resistance training. I have noted below but I believe adding clarity would be of value for future reader. 

Specific Comments

Abstract

-       line 16-17 -musculoskeletal injuries due to exercise? What form of exercise are you referring to?

Introduction.

- line 53- Exercise injury or exercise induced injury? You refer in remaining manuscript as exercise injury 

-Consider definition of an ‘exercise injury’ – are you referring to resistance training? Or a combination of modes? 

Methods

- how was the final number of participants determined from those invited to participate? 

- there were large differences between AT vs PT and the # of encounters- would subgroup analyses be considered? From table 1 potential subgroups would be AT vs. PT and volume of FF patient encounters 

Results

-I found the quotes valuable- Consider adding a supplement table to present direct quotes and themes found

-However, I would also point out several were challenging to follow, possibly re-consider which quotes to include or how much of the quote in order to maintain clarity

-Table 6- there is a random ‘a’ within row 3

Discussion

References

-# 5 and 12- are these the doi for the proceeding references? 

Round 2

Reviewer 1 Report

Comments and Suggestions for Authors

I have no further edits or comments.